# The Relationship of Family Functioning and Suicidal Ideation among Adolescents: The Mediating Role of Defeat and the Moderating Role of Meaning in Life

**DOI:** 10.3390/ijerph192315895

**Published:** 2022-11-29

**Authors:** Qin Yang, Yi-Qiu Hu, Zi-Hao Zeng, Shuang-Jin Liu, Tong Wu, Gang-Huai Zhang

**Affiliations:** 1School of Education Science, Hunan Normal University, Changsha 410081, China; 2School of Pre-School Education, Changsha Normal University, Changsha 410100, China; 3School of Psychology, Hainan Normal University, Haikou 571158, China

**Keywords:** adolescents, family functioning, suicidal ideation, defeat, meaning in life

## Abstract

Objective: To investigate the relationship between family functioning and suicidal ideation among adolescents. Method: A total of 4515 junior and senior high school students were assessed using the Family APGAR, the Depressive Symptom Index-Suicidality Subscale, the Defeat Scale, and the Chinese Meaning in Life Questionnaire. Results: This study found pairwise correlations between suicidal ideation, family functioning, defeat, and meaning in life. Specifically, family functioning was an influencing factor of adolescent suicidal ideation, and defeat was a mediator of the relationship between family functioning and adolescent suicidal ideation; meaning in life was found to be a moderator of the first half of the mediation process by defeat, that is, it moderated the influence of family functioning on adolescent defeat. Conclusions: This study demonstrated that the relationship between family functioning and adolescent suicidal ideation, as well as the influence of defeat and meaning in life on this relationship, constituted a moderated intermediary model. This finding has both theoretical and practical value for the implementation of a psychosocial model of adolescent suicide prevention and intervention.

## 1. Introduction

Suicidal ideation refers to the process in which individuals form thoughts and content about ending their own lives without taking concrete actions [1]. It is more common among adolescents, with a reported detection rate of 19.3% to 39.4% in the United States, Germany, etc. [2], and 11.6% to 24.7% over the past 20 years in China [3,4]. In the last two years, the global COVID-19 pandemic has also had an impact on the mental health of adolescents [5], resulting in an increased incidence of suicidal ideation and behaviors [6,7]. Suicidal ideation is an important indicator of mental health issues [8,9], as well as the most sensitive predictor of suicidal attempts and behaviors [8,10,11]. Evidence has suggested that the development of suicidal ideation to attempts results from complex interactions between personal and environmental factors [12]. Therefore, a better understanding of the influencing factors [13] and the improvement of effects of intervention strategies are urgently needed to prevent adolescents’ suicide, especially when the COVID-19 pandemic is predicted to become endemic [14].

### 1.1. The Relationship between Family Functioning and Suicidal Ideation

The Ecological System Theory [15] holds that individual development is affected by a series of interactive environmental systems, among which family was the most important microenvironmental system that affects adolescent social adaptation as well as physical development. Olson put forward the concept of family functioning, which reflects the characteristics of the family system [16]. According to Olson, family functioning can serve as an important index in the evaluation of the operation, relationship between family members, family adaptation, and other aspects of the family system [16], reflecting the emotional bond between family members, family rules, communication within family, and the effectiveness of coping with external events [16]. Thus, family functioning is an underlying variable that affects the social adaptation and psychological development of adolescents [17,18]. Some recent studies [12,16] revealed that better family functioning was associated with higher levels of physical and mental health of family members, as well as better social adaptation [17,19], while family dysfunction was associated with more explicit and implicit problems in adolescents. For example, the literature has demonstrated a direct and negative correlation between family functioning and the incidence of depression, anxiety, suicidal ideation, etc., in adolescents [9]. Furthermore, numerous studies have shown that parental rejection is associated with self-destructive thoughts and behaviors in adolescents [20,21], lower levels of parental care were significantly associated with higher levels of suicidal ideation [22], and parental rejection/neglect was associated with a higher risk of adolescents’ suicidal attempts [23]. Studies have also reported that poor family functioning was predictive of higher levels of suicidal ideation among junior high school students [24], and that higher family closeness, adjustment, and support could reduce the incidence of suicidal ideation [25]. A cross-sectional study of adolescents aged 11 to 18 years in two Chinese developed cities, Hong Kong and Shanghai, confirmed that family functioning significantly predicted suicidal ideation in this group [26], but the mechanism of how family functioning develops as a risk factor for suicidal ideation and behavior remains to be explored in greater depth [27].

### 1.2. The Mediating Effect of Defeat

The Ecological System Theory emphasizes the interaction between systems and individuals as well as the effect of this interaction on individual development [15]. Relevant studies supported that individual factors are also important variables affecting suicidal ideation. Some researchers have focused on the variable of defeat, defining it as the individual perception of failure in social competition, loss, and decline in social status [28]. The Interpersonal Psychological Theory of Suicide (IPTS) [29], the Schematic Appraisals Model of Suicide (SAMS) [30], and the Integrated Suicide Model [31] all suggest that defeat is an important risk variable for predicting individual suicidal ideation. Among the many theoretical models, the most representative and widely accepted theoretical model—the Integrated Motivational-Volitional Model of Suicidal Behavior [10,12,32]—integrates the theoretical perspectives and influential variables from various previous models. The model conceptualizes suicide as a planned action performed by an individual rather than the result of a mental disorder [12], and defeat is the core predictive variable in the stage of suicidal ideation formation [10]. In the IMV Model, suicide is divided into the pre-motivational stage, motivational stage, and volitional stage. The pre-motivational stage is based on the diathesis-stress model, where diathesis refers to biological or genetic vulnerabilities, while stress consists mainly of environmental factors (e.g., deprivation) and negative life events (e.g., relationship breakdown). The motivational stage refers to the formation of suicidal ideation; during this stage, individuals with a psychological base and stressful life may experience defeat, which could result in entrapment and hopelessness if the sense of defeat is unrelieved or there is a lack of social support, and further predicts suicidal ideation as the solution of life’s problems [10,33,34]. In the volitional stage, suicidal ideation develops into suicidal attempt or behaviors, with the suicidal ability and acquisition of tools being moderators. The theoretical hypothesis of this model has been supported by numerous studies. A meta-analysis by Taylor et al. showed that there is a stable and reliable correlation between defeat and suicidal ideation, with individuals experiencing high levels of defeat possessing higher levels of suicidal ideation over a 12-month period [35]. In a 4-year follow-up study on patients hospitalized after suicidal attempts [12], defeat was found to be moderately associated with suicidal ideation and predictive of subsequent attempts. Multiple studies on college students with suicidal ideation [36], bipolar disorder [37], schizophrenia [35], and post-traumatic stress disorder [35] have concluded that defeat could predict suicidal ideation and explain the greater variation in suicide risk.

According to one of the theoretical foundations of the IMV model, the Interpersonal Theory of Suicide [38], individuals perceiving themselves as burdensome to others and lacking social support will increase the possibility of individuals from defeat to intrapment and suicidal thoughts. Attachment theory [39,40] also believes that childhood exposure to destructive family environments (e.g., parental abuse, neglect, inconsistency, and ineffective parenting) can lead into the development of insecure attachment styles (i.e., anxiety and avoidance), as well as negative perceptions of themselves and others, such as feelings of being a burden to others, feelings of isolation, or lack of social support, which are associated with systemic interpersonal dysfunction throughout their lives [41]. In addition, more studies have attested that stress in the early-life environment (such as parental neglect) can affect individuals’ emotions and cognitive ability [42,43] and determine the likelihood of them developing psychological disorders and mental crises. Studies also showed that stressors in the early-life environment (such as parental neglect) had a negative effect on individuals’ emotions and cognitive abilities [42]. Zortea et al. [27] attested that poor parent–child attachment was associated with increased suicidal thoughts and behaviors through the mediation of feelings of defeat and entrapment, while being respected and cared for in the family could improve the resilience of individual psychological trauma. According to Van Petegem et al. [44], higher levels of overprotective parenting as rated by adolescents were associated with a higher risk of their experiencing maladjustment and defeat. Li et al. [45] discovered that poor family functioning might lead individuals to develop negative stress-coping styles and, thus, become more susceptible to suicidal ideation. In the light of the above findings, defeat may be an important mediator of the effect of family functioning on adolescent suicidal ideation. In addition, with suicide being a social and cultural phenomenon, the adaptability of the IMV model to adolescents in more representative cities of Chinese adolescents also needs to be tested.

### 1.3. The Moderating Effect of Meaning in Life

With the rise of positive psychology in recent years, protective factors of suicidal ideation have gradually become the focus of research [46]. As proposed by Frankl [47], “Meaning in Life” has rapidly become a hot spot in the field of positive psychology. Meaning in life refers to the degree to which individuals comprehend, understand, and perceive the purpose, mission, and goal of their life [48]. According to Frankl, meaning in life is a power to influence one’s work, creation, and ability to endure hardship [49]. Studies have also shown that meaning in life is significantly but negatively associated with suicidal ideation [50], and can effectively reduce the risk of suicide among adolescents [51]. Meaning in life was also found to be one of the protective factors against impulsive suicidal ideation or behaviors [52]. According to the IMV Model theory, diathesis (e.g., personality traits and cognitive style) is an important influencing factor in the pre-suicide motivational stage; it makes individuals suffering from adverse situations or life events realize that there are other choices in life, which buffers the emergence of defeat and suicide thoughts and facilitates the production of positive thoughts about the future [32]. It was found that greater meaning in life (and related construction) was associated with a better self-control of cognition and emotions among adolescents and adults, which was more conducive to a flexible response to adverse situations and reduces the occurrence of defeat. Moreover, a survey showed that the interaction between meaning in life and family intimacy was predictive of the mental health of individuals [53], and that the interaction between meaning in life and family functioning was predictive of the risk of suicide [54]. 

In addition, the theoretical perspective derived from Erikson suggested that developing a sense of meaning in life is an important task for adolescents [55,56]. The search for meaning in life could be a difficult period. However, once established, it can have a positive and lasting impact on all aspects of their lives. However, few studies have examined this issue so far. Therefore, under the theoretical framework of the IMV Model, when adolescents encounter a dysfunctional family situation, it needs to be explored whether a higher level of meaning in life can function as a resource for adolescents to strengthen their self-control and regulation [56] and serve as a buffering factor against the emergence of defeat, thereby reducing risks of suicide motivation.

## 2. Participants and Measures

### 2.1. Participants

The data of this study came from a large-scale domestic epidemic survey, and the subjects were junior and senior high school students in Hunan, Guangdong, Jiangxi, and Anhui Province in China. The survey was conducted from March to July 2022 and administered in class settings. Informed consent was obtained from all the subjects’ teachers and parents before the investigation initiated. The main examiners were graduate students majoring in psychology from Hunan Normal University, and they received training beforehand to ensure consistency. During the survey, the students (i.e., participants) were informed that they could submit the questionnaire at any time if feeling unwell. The whole questionnaire took about 40~60 min to complete. All the information was kept confidential. All materials and procedures of this investigation were approved by the ethics review committee of the university. A total of 5000 questionnaires were distributed, and 4515 valid questionnaires were returned after excluding invalid responses, a recovery efficiency of 90.3%. Among the participants who submitted valid responses, there were 2268 boys (50.2%) and 2247 girls (49.8%), with a mean age of 15.24 years (SD = 1.66); 1070 (23.7%) were junior high school grade 1 students, 963 (21.3%) were junior high school grade 2 students, 1370 (30.3%) were senior high school grade 1 students, and 1112 (24.6%) were senior high school grade 2 students. Students in the third year (i.e., the final year) of junior and senior high school were not selected as participants, as they were facing the High School Entrance Exam and College Entrance Exam.

### 2.2. Measures

#### 2.2.1. The Depressive Symptom Index-Suicidality Subscale (DSI-SS)

The Depressive Symptom Index-Suicidality Subscale (DSI-SS), developed by Joiner, Pfaff, and Acres [13], is used to investigate the frequency and intensity of suicidal ideation and impulse over a two-week period as a brief screening device for suicide ideation in general health settings. The scale has been shown to be effective in the assessment of the severity of suicidal ideation among adolescents [13,57,58,59]. The scale consists of 4 items, and the score of each item ranges from 0 to 3 (e.g., from “I have no suicidal thoughts” to “I always have suicidal thoughts”), and the total score ranges from 0 to 12, with higher scores indicating a higher tendency of suicidal ideation. In clinical practice, 2 points are often used as the lowest cut-off point for suicidal ideation in demographic samples [60]. Before the use of the scale, a doctoral student majoring in English was invited to translate it into Chinese, and an English teacher was asked to proofread the translated manuscript. The results of exploratory factor analysis showed that KMO = 0.843 and spherical test *p* < 0.05. Confirmatory factor analysis showed that the fitting index of the model was good, χ^2^/df = 22.31, *p* < 0.001, CFI = 0.997, TLIQ = 0.984, RMSEA = 0.069, and the structure validity was good. The Cronbach’s α coefficient of the scale was 0.917 in this study. 

#### 2.2.2. The Family APGAR

The Family APGAR was compiled by Smilkstein [61] and introduced to China by Lv et al. [62]. It is a tool to measure individual satisfaction with their family functioning in a subjective way. The questionnaire includes aspects of family adaptation, partnership, growth, affection, and resolve. Each item was rated on a 3-point scale (0 = almost rarely, 1 = sometimes, and 2 = often), and the total score of the five items ranged from 0 to 10. The scores of 0–3 indicated severe family dysfunction, 4–6 indicated moderate family dysfunction, and 7–10 indicated good family functioning. The Cronbach’s α coefficient of the scale was 0.866 in this study.

#### 2.2.3. The Defeat Scale

The Defeat Scale (DS), compiled by Gilbert and Allan [28], was used to assess individual views on social status decline and failure in competitions in the past week. The scale consists of 16 items, and each item was rated on a 5-point scale, with 1 indicating “never” and 5 indicating “always”; higher total scores indicated a stronger sense of defeat. In this study, we used the Chinese version of the scale revised by Tang et al. [63]. The Cronbach’s α coefficient of the scale was 0.885 in this study.

#### 2.2.4. Chinese Meaning in Life Questionnaire

The Meaning in Life Questionnaire (MLQ) was compiled by Steger et al. [48]. There are 10 items in total, including 2 factors, i.e., Presence of Meaning (PM, the degree of feelings about whether one’s life is meaningful, emphasizing the result) and Search of Meaning (SM, the degree of active pursuit of the meaning in life, emphasizing the process). Each item was rated on a 7-point Likert scale (from 1 to 7). The Chinese version of the questionnaire (C-MLQ) was revised by Wang and Dai [64], and it was demonstrated that the scale had good reliability and validity. Steger et al. [48] showed that there was a high correlation between the two subscales in Asian countries (such as Japan and China) under the cultural background of collectivism, so this study used the method of packaging to calculate the total score, namely the higher the total score, the higher the overall level of meaning in life. The Cronbach’s α coefficient of the scale was 0.813 in this study.

### 2.3. Statistical Analysis

SPSS 26.0 was used for descriptive and correlation analyses, and the PROCESS3.5 macro program plug-in developed by Hayes [65] was used to evaluate the moderated mediation effect model. All variables were standardized, while gender and school year were encoded using virtual variables. The data were tested for significance of effects using a bias-corrected percentile bootstrap method with repeated sampling for 5000 times and 95% confidence intervals calculated, in order to obtain standard errors of parameter estimates [65,66]. In addition, the data in this study were self-reported by participants; thus, common method biases could not be all-together avoided. Therefore, 5 items of family functioning, 4 items of suicidal ideation, 16 items of defeat, and 10 items of meaning in life were packaged. Exploratory factor analysis was performed using the Harman single factor test, and unrotated principal component factor analysis was used for the test. The characteristic root of 6 factors was greater than 1, and the explained variance ratio of the first factor was 26.94% (lower than the critical value of 40%), suggesting no serious common method biases.

### 2.4. Research Hypotheses

By introducing defeat as a mediating variable and meaning in life as a moderating variable, this study aimed to investigate the relationship between family functioning and suicidal ideation in Chinese adolescents. The research hypotheses were as follows: 

**H1.** 
*Family functioning might be a significant predictor of adolescent suicidal ideation.*


**H2.** 
*Defeat might not only directly affect adolescent suicidal ideation, but also plays a mediating role between family functioning and suicidal ideation in adolescents.*


**H3.** 
*Meaning in life might not only directly moderate the relationship between family functioning and suicidal ideation, but also the first half of the intermediary path of “family functioning → defeat → suicidal ideation”.*


## 3. Results

### 3.1. Descriptive Statistics and Correlation Analysis

Based on a score of two or more for SI [60], 1767 adolescents with suicidal ideation in the past two weeks were detected, accounting for 39.1% of the total number. Comparing means and *t*-tests showed a significant difference in scores of suicidal ideations between boys and girls (*t* = −7.14, *p* < 0.001); girls’ (2.48 ± 3.20) were significantly higher than boys’ (1.83 ± 2.88). In addition, grade differences were extremely significant (*t* = 9.57, *p* < 0.001), with the junior high school group (2.64 ± 3.41) significantly higher than the senior high school group (1.76 ± 2.72). Correlation analysis (see Table 1) showed that suicidal ideation was negatively correlated with family functioning and meaning in life (*p* < 0.001), and positively correlated with defeat (*p* < 0.001); family functioning was found negatively correlated with defeat (*p* < 0.001) and positively correlated with meaning in life (*p* < 0.01); defeat was found negatively correlated with meaning in life.

### 3.2. Family Functioning and Suicidal Ideation in Adolescents: A Moderated Mediating Effect Test

The mediating effect of defeat was first tested (see Table 2, Model 1 and Model 2). The results showed that family functioning had a direct negative effect on suicidal ideation (*β*= −0.34, *p* < 0.001). After adding the mediating variable, defeat, family functioning was found to have a direct negative effect on defeat (*β* = −0.33, *p* < 0.001), and defeat was shown to have a direct positive effect on suicidal ideation (*β* = 0.38, *p* < 0.001), while family functioning still had a significant and negative effect on suicidal ideation (*β* = −0.20, *p* < 0.001). The findings demonstrated that defeat played an intermediary role in the relationship between family functioning and adolescent suicidal ideation, which supported hypotheses 1 and 2 of this study. The Bootstrap test for bias correction showed that the indirect effect value of the mediating effect was −0.14, accounting for 41.2% of the total effect (−0.34), and the direct effect value was −0.20, accounting for 58.8% of the total effect (−0.34).

Further analysis of the moderating effect of meaning in life showed that the interaction between meaning in life and family functioning only had a significant effect on defeat (*β* = 0.07, *p* < 0.001; see Table 2, Model 2), but not on suicidal ideation (*β* = 0.02, *p* > 0.05; see Table 2, Model 3). In addition, the indirect effect of family functioning on suicidal ideation also gradually diminished at three different levels of meaning in life (*p* < 0.001, see Table 3). This suggested that, compared with adolescents with greater meaning in life, the mediating effect of defeat of adolescents with low meaning in life on the relationship between family functioning and suicidal ideation significantly reduced, i.e., meaning in life regulated the mediating effect. Thus, the moderated mediation model was established (Index = 0.03, SE = 0.01, 95%CI = [0.02, 0.04], see Table 3). The above findings suggested that the mediating effect of meaning in life on the relationship of defeat on family functioning and suicidal ideation occurred in the first half of the model, rather than the direct path. It could be inferred that there were significant moderated mediating effects in the model, so part of hypothesis 3 of this study was supported.

In addition, we used simple slope analysis to further understand the regulatory mechanisms of meaning in life (see Figure 1). The results showed that the difference of adolescent defeat with different levels of family functioning was more significant when the level of meaning in life was low (*β*_simple_ = −0.40, *t* = −22.26, *p* < 0.001), while this difference was slightly lower at a high level of meaning in life (*β*_simple_ = −0.26, *t* = −14.52, *p* < 0.001). The results of the analysis showed that the predictive effect of family functioning on defeat decreased significantly as the level of meaning in life increased. Therefore, the amount of the mediated effect of defeat tended to decrease (Table 3), that is, as the level of meaning in life increased, family functioning was less likely to directly induce adolescent defeat. In conclusion, this model supported a moderated mediation model in which meaning in life was a moderator of family functioning, affecting adolescent suicidal ideation through defeat.

## 4. Discussions

### 4.1. Current Status of Suicidal Ideation among Adolescents

Adolescence is a time of “storm and stress”, when the personality has not yet matured and the level of suicidal ideation and behaviors can be relatively high under extreme stress [67]; thus, adolescents have been intensively studied in suicide research. Surveys in some Western countries have suggested that the prevalence of suicidal ideation was approximately 15–30.8% among individuals under the age of 18 [68], which decreased sharply to 3.7–14.3% after they became adults [69]. This study, based on a multi-provincial large-scale survey, also showed that the detection rate of suicidal ideation among adolescents in secondary schools was as high as 39.1%, higher than previous findings [2,3,4]. To some extent, these results reflected an elevated risk of psychological crisis among adolescents in secondary schools when the COVID-19 pandemic occurred [70,71]. In addition, the suicidal ideation of girls and junior high school students was significantly higher than that of boys and high school students, which was consistent with the conclusions of some domestic studies [24,72]. In junior high schools, boys and girls are both in adolescence going though rapid physical and mental development, but girls bear more pressure from physical development, school adaptation, and other aspects of pressure than boys, adding to the susceptibility characteristics of girls themselves, making them more susceptible to such risks, which can lead to negative emotions such as depression and suicidal ideation. In addition, the subjects in this study were all from key junior high schools, with a heavier academic burden and higher expectations from parents than ordinary schools, which can also lead to fatigue, aggravation of mental health problems, and a sharp increase in the risk of related psychological crises such as self-injury and suicidal ideation [70]. For example, studies have found that the mental health of Chinese junior high school students was deteriorating, especially in the central and western regions [19], which necessitates special attention from schools and families.

### 4.2. Relationship between Family Functioning and Adolescent Suicidal Ideation

This study found that family functioning was not only correlated with adolescent suicidal ideation, but also negatively affected suicidal ideation. A study involving 3178 Hispanic adolescents in grades 9–12 found that teenagers brought up in a disharmonious family and lacking parental love and care were 2.6–5 times more likely to have suicidal ideation [73]. Previous studies also demonstrated that the occurrence of suicidal ideation was positively correlated with family discord, poor family environment, family rigidity, family conflicts, and poor adaptability [74], and negatively associated with family intimacy, emotional expression, and organization [75]. Low levels of family cohesion and support and high levels of parent–adolescent conflict increased the risk of adolescent depression and suicidal ideation [7,76], whereas higher levels of family closeness, adjustment, cohesion, and family support reduced adolescent suicidal ideation. Our study further corroborated the above findings. According to the Ecological System Theory, family is the most important microenvironment system and the first “protective barrier” for the physical and mental health of adolescents. As adolescents are in a critical period of physical and mental development and are faced with a variety of adaptive challenges [77], their families are important in supporting their physical and mental health [78]. As collectivistic cultures (e.g., Chinese) lend more emphasis to interpersonal relationships, poor family functioning may be more closely linked to suicidal ideation in such cultures than in individualistic ones (e.g., Western). Meanwhile, collectivism attaches great importance to family relations. Families with a harmonious atmosphere, high-quality communication, and mutual concern can better emotionally support, guide, and help, which all play important roles in preventing adolescent suicidal ideation [79,80]. In addition, more studies have confirmed that family functioning was a significant predictor of suicide in Chinese culture [26].

### 4.3. The Mediating Role of Defeat

This study also examined the influence of defeat on the relation between family functioning and adolescent suicidal ideation. The results showed that defeat not only positively affected adolescent suicidal ideation, but also played an intermediary role in the relation between family functioning and adolescent suicidal ideation; this demonstrated that defeat might be a risk factor for adolescent suicidal ideation and that adolescents with poor family functioning, weak parent–child emotional connection, and less support from families were more likely to feel defeated, which increased the risk of suicidal ideation. A study involving 1239 junior high school students in China also demonstrated that defeat was predictive of suicidal ideation, directly and indirectly, through the feelings of entrapment. A study on 730 adults found that defeat played an intermediary role in negative educational styles and suicidal ideation. Both the diathesis-stress model and integrated motivational-volitional model suggest that psychological disorders (e.g., depression, anxiety, self-injury, and suicide) and obstacles are the results of the joint effect of external pressure and internal “susceptibility”. These findings indicated that adverse environmental factors (poor family functioning) could contribute to adolescent suicidal ideation through individual cognitive factors (defeat experience), which was consistent with the conclusion of previous studies [35] and supported the stage theory of suicide in the MIV model, which stated that suicide was a consciously planned and implemented behavior based on the individual condition and the environment, rather than merely a manifestation or product of psychological issues or mental disorders [81].

### 4.4. The Moderating Effect of Meaning in Life

Studies have demonstrated that meaning in life not only enhanced happiness, but also helped individuals cope with psychological risks in particular stages of human development. Pursuing a sense of purpose in life could help reduce the negative impact of stressful situations, enhance the quality and value of life, and reduce the risk of psychological disorders and crises such as suicide [82]. A study showed that college students with a good family atmosphere and functioning were more likely to actively pursue a meaning in life and constantly accept and improve themselves, which contributed to their higher sense of meaning in life and a lower risk of suicide [68]. Through examining the influence of meaning in life on middle and high school students by including it as a regulatory variable, this study showed that meaning in life had a significant regulatory effect on the first half of the mediating model of the effect of defeat on the relation between family functioning and suicidal ideation. The result further supported the role of meaning in life as an important psychological resource in buffering the influence of poor family functioning on adolescents’ feelings of defeat. The impact family functioning had on defeat decreases with the pursuit of meaning in life, suggesting that meaning in life was a protective factor, especially for adolescents with poor family functioning. The result also corroborated the theoretical framework of the IMV Model. The pursuit of meaning in life enabled adolescents to maintain their goals and beliefs and cherish the value of life, which helped buffer the influence of poor family functioning on defeat and subsequently reduced the occurrence of suicidal ideation [10].

## 5. Practical Significance

Based on the Ecological System Theory and the IMV Model, this study confirmed the research hypothesis that the influence of family functioning on adolescent suicidal ideation was a moderated mediation model, adding evidence to the application of the IMV model among Chinese adolescents. As the study data were obtained from a large prevalence survey across four provinces and cities in China, the finding is representative and has certain implications for a systematic understanding of how family functioning affects adolescent suicidal ideation, and for the development of effective suicide prevention measures for Chinese adolescents by educational authorities and mental health agencies.

First, this study further supports that a comprehensive focus on psychosocial risk factors is critical for the early prevention of adolescent suicidal behavior. For example, adolescents who have poor family functioning and experience high levels of defeat may be highly susceptible to suicide attempts, and external adverse environments can increase adolescents’ risk of suicidal crisis through individual cognitive factors. Suicide is a complex phenomenon involving many risk factors and synergistic effects among all relevant variables may be important [32]. Hence, suicide prevention efforts for adolescents support an integrated approach that strengthens adolescent family functioning and promotes positive self-perceptions among adolescents, rather than focusing only on high-risk adolescent groups and their single risk factors. This also provides a reference mechanism for the education sector and health care providers. A study has shown that adolescents hospitalized for suicide attempts benefitted from tailored preventive measures based on psychosocial factors [83]. According to Calear et al. [84], 86% of individual and parent/family intervention approaches were significantly effective in the management of suicidal ideation and attempts in adolescents. In particular, the characteristic of Chinese culture is family-centered. Family intervention should be included as a special measure in the overall intervention program, so that families can reduce adverse effects and play a protective role in the process of adolescent suicide prevention.

In addition, this study also confirms that meaning in life would be an important protective factor in adolescent suicide prevention. Given that adolescence is a critical period in identity development [55], adolescents with healthy worldviews and emotions (e.g., positive meaning in life) were likely to have a positive evaluation of themselves and a better perception of meaning in life, which could serve as a favorable entry point into adolescents [85]. Meaning therapy proposed by Frankl et al. may be a useful strategy for those who grew up in dysfunctional families [86]. It can guide adolescents to tap into sustainable, transcendent meaning to fill an inner void, enabling them to see themselves in a new and positive light, as well as to pursue life in a responsible, purposeful, and hopeful way [47]. Meanwhile, authentic and therapeutic relationships (e.g., positive parent–child relationships and teacher–student relationships) can help adolescents increase their sense of connection and belonging to others and society, thus effectively preventing their risk of depression and suicide, and promoting essential changes in them.

## 6. Limitations and Future Implications

This study has several drawbacks. First, the study is cross-sectional, which precluded us from examining the causal relationship between independent and dependent variables; longitudinal investigations and cross-lagged analyses can be conducted in future works. Secondly, only gender and grade were included as controlled variables, with no in-depth discussion of their influence on the relation between independent and dependent variables; thus, demographic factors such as gender, grade, age, family structure, and household economic status can be used in future studies as regulatory variables to explore their roles. Thirdly, data collected in this study were mostly self-reported, making common method biases unavoidable despite the multiple-source and large-scale data collection. Thus, in future studies, measurement control needs to be bettered, e.g., using the Implicit Association test or adding social desirability variables. Finally, given that the core variables of the motivational stage in the IMV Model also included entrapment and hopelessness, future studies can incorporate them as mediating variables to further validate their role and mechanism in the developmental pathway of suicide among Chinese adolescents.

## 7. Conclusions

In conclusion, this study demonstrated that the relationship between family functioning and adolescent suicidal ideation, as well as the influence of defeat and meaning in life on this relationship, constituted a moderated intermediary model. This finding has both theoretical and practical value for the implementation of a psychosocial model of adolescent suicide prevention and intervention.

## Figures and Tables

**Figure 1 ijerph-19-15895-f001:**
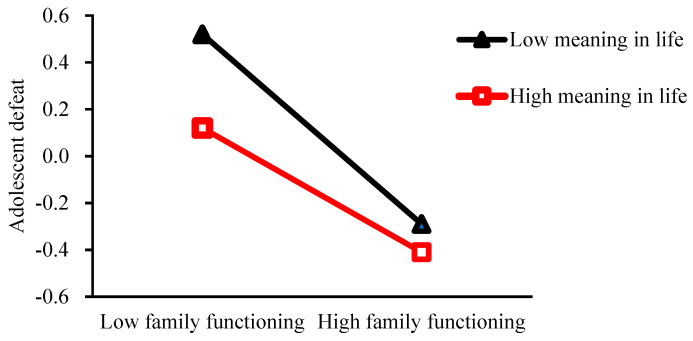
Simple slope trajectory of the moderating role of meaning in life on the relationship between family functioning and adolescent defeat.

**Table 1 ijerph-19-15895-t001:** Descriptive statistics and correlation analysis results with regard to family functioning, defeat, meaning in life, and suicidal ideation.

Category	M	SD	1	2	3	4	5	6
1.Family functioning	5.71	2.84						
2.Defeat	35.11	10.94	−0.365 ***					
3.Meaning in life	47.62	10.68	0.193 **	−0.200 ***				
4.Suicidal ideation	2.16	3.08	−0.336 ***	−0.431 ***	−0.131 ***			
5.Gender	0.50	0.50						
6.Grade	0.45	0.50						

Note: *n* = 4515; Gender: male = 1, female = 0; Grade: junior high school group = 1, senior high school group = 0; ** *p* < 0.01, *** *p* < 0.001.

**Table 2 ijerph-19-15895-t002:** Testing of the moderated mediating effect.

Variable	Model 1 (Dependent Variable: Y)				Model 2 (Dependent Variable: M)				Model 3 (Cependent Variable: Y)			
	*β*	*SE*	*t*	95%CI	*β*	*SE*	*T*	95%CI	*β*	*SE*	*t*	95%CI
Gender	0.21	0.03	7.62 ***	[0.16, 0.26]	0.17	0.03	0.61	[−0.04, 0.07]	0.21	0.03	8.16 ***	[0.16, 0.23]
Grade	0.33	0.03	11.98 ***	[0.28, 0.39]	−0.23	0.03	−8.54 ***	[−0.29, −0.18]	0.42	0.03	16.22 ***	[0.37, 0.47]
X	−0.34	0.01	−24.68 ***	[−0.37, −0.31]	−0.33	0.01	−23.98 ***	[−0.36, −0.31]	−0.20	0.01	−14.54 ***	[−0.23, −0.17]
W					−0.13	0.01	−9.39 ***	[−0.16, −0.10]	−0.02	0.01	−1.58	[−0.05, 0.01]
X × W					0.07	0.01	6.19 ***	[0.05, 0.09]	0.02	0.01	1.60	[−0.00, 0.04]
M									0.38	0.01	27.20 ***	[0.35, 0.41]
M × W												
*R* ^2^	0.15				0.17				0.28			
*F*	262.85 ***				184.08 ***				285.76 ***			

Note: Gender: male = 1, female = 0; Grade: junior high school group = 1, high school group = 0; X = family functioning; M = defeat; W = meaning in life; Y = suicidal ideation. *** *p* < 0.001.

**Table 3 ijerph-19-15895-t003:** Direct and mediating effects at different levels of meaning in life.

Type of Effect	Meaning in Life	*Β*	BootSE	BootLL95%CI	BootUL95%CI
The direct effect	M − 1SD	−0.22	0.02	−0.25	−0.18
	M	−0.20	0.01	−0.23	−0.17
	M + 1SD	−0.18	0.02	−0.22	−0.15
The mediating effect of defeat	M − 1SD	−0.15	0.01	−0.18	−0.13
	M	−0.13	0.01	−0.14	−0.11
	M + 1SD	−0.10	0.01	−0.12	−0.08
	Index	0.03	0.01	0.02	0.04

## Data Availability

Data from this study have been made public as required and can be accessed at https://osf.io/63ytz/ (accessed on 20 November 2022).

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
