# Peer review of "The Relationship of Family Functioning and Suicidal Ideation among Adolescents: The Mediating Role of Defeat and the Moderating Role of Meaning in Life"

_ijerph, 2022, doi:10.3390/ijerph192315895_

Round 1

Reviewer 1 Report

Thank you for the opportunity to review this manuscript on the role of family functioning and suicidal ideation in adolescents in Chinese adolescents.

I have the following suggestions:

Title: strengthen the title by making it richer in information to include 'defeat' and 'meaning in life' as the association between family functioning and suicidal ideation in adolescents has already been established

1. Introduction, line 69, the statement that the best model is O'Connor is based only on O'Connor citations (refs 12, 14, 37). The authors of this study should provide readers an overview of the numerous models that exist by citing for example Diaz-Olivan DOI: 10.1016/j.ejpsy.2021.02.002 who provide an overivew of the models and provide their own rationale for why they chose the IMV, as well as cite Leung https://doi.org/10.1007/s10597-015-9920-2 who have a model specific to ideation.

1. Introduction, line 151, should be a new paragraph as does not belong to subheading 1.3

1. Introduction, lines 154-155 Hypothesis 1, this has already been tested, see: Leung, C.L.K., Kwok, S.Y.C.L. & Ling, C.C.Y. An Integrated Model of Suicidal Ideation in Transcultural Populations of Chinese Adolescents. Community Ment Health J 52, 574–581 (2016). https://doi.org/10.1007/s10597-015-9920-2. Therefore, please adjust to provide a rationale for why you are testing this in this population and cite this paper in the intro

2. Methods as heading missing, which is standard; remove 2. Subjects and measures

4. Discussion - it is missing how the results are useful theoretically and practically, as cited in the conclusion. Only mention is line 437 section limitations: 'these findings are helpful for education departments and mental health institutions to devise effective suicide prevention plans for Chinese adolescents' which is not specific or informative enough.

Author Response

Response to Reviewer 2 Comments

Dear Reviewer:

We are very grateful to your comments for the manuscript. Those comments are all valuable and very helpful for revising and improving our paper, as well as the important guiding significance to our researches. We have studied comments carefully and have made correction which we hope meet with approval. Revised portion are marked in red in the paper. The main corrections in the paper and the responds to the reviewer’s comments are as flowing:

Responds to the Reviewer’s comments:

Piont 1: Title: strengthen the title by making it richer in information to include 'defeat' and 'meaning in life' as the association between family functioning and suicidal ideation in adolescents has already been established.

Response 1: As suggested by the Reviewer, we have revised the title of the article to “The Relationship of Family Functioning and Suicidal Ideation among Adolescents: the Mediating Role of Defeat and the Moderating Role of Meaning in Life ”. The revised title already clearly describes the mediating and moderating variables of our study.

Piont 2: 1 Introduction, line 69, the statement that the best model is O'Connor is based only on O'Connor citations (refs 12, 14, 37). The authors of this study should provide readers an overview of the numerous models that exist by citing for example Diaz-Olivan DOI: 10.1016/j.ejpsy.2021.02.002 who provide an overivew of the models and provide their own rationale for why they chose the IMV, as well as cite Leung https://doi.org/10.1007/s10597-015-9920-2 who have a model specific to ideation.

Response 2: We are very sorry for our negligence of not listing and presenting the multiple theoretical models of suicide in detail in our research paper, but only presenting the most representative and widely accepted theoretical model—the Integrated Motivational-Volitional Model. Thanks to the Reviewer’s guidance and recommended literature, we have been able to add important references to our research. We have carefully studied the literature you recommended and other relevant literature. Thus, we have made the following adjustments to this section:  

Line 63-67, “A cross-sectional study of adolescents aged 11 to 18 years in two Chinese developed cities, Hong Kong and Shanghai, confirmed that family functioning significantly predicted suicidal ideation in this group [1]” was added.

Line 74-83, “The Interpersonal Psychological Theory of Suicide (IPTS) [2], the Schematic Appraisals Model of Suicide (SAMS) [3] and the Integrated Suicide Model [4] all suggest that defeat is one of the important risk variables for predicting individual suicidal ideation. Among the many theoretical models, the most representative and widely accepted theoretical model, the Integrated Motivational-Volitional Model of Suicidal Behavior [5-7] integrats the theoretical perspectives and influential variables from various previous models. The model conceptualizes suicide as a planned action performed by an individual rather than the result of a mental disorder [5], as well as defeat is the core predictive variable in the stage of suicidal ideation formation [6]” was added.

The main references are listed below:

[1]Leung CL, Kwok SY, Ling CC. An Integrated Model of Suicidal Ideation in Transcultural Populations of Chinese Adolescents. Community Ment Health J. 2016; 52(5):574-81. http://doi: 10.1007/s10597-015-9920-2. Epub 2015 Aug 27 PMID: 26308835

[2]Joiner TE. Why people die by suicide. Cambridge: Harvard University Press, 2005

[3]Johnson J, Gooding P, Tarrier N. Suicide risk in schizophrenia: explanatory models and clinical implications, The Schematic Appraisal Model of Suicide (SAMS). Psychol Psychother. 2008; 81:55-77. http://doi:10.1348/147608307X244996 PMID:17919360

[4]Taylor P, Gooding P, Wood A, Tarrier N. The role of defeat and entrapment in depression, anxiety, and suicide. Psychol Bull. 2011; 137:391-420. http://doi:10.1037/a0022935 PMID: 21443319

[5]Díaz-Oliván A, Porras-Segovia ML, Barrigón L, Jiménez-Muñoz E, Baca-García. Theoretical models of suicidal behaviour: A systematic review and narrative synthesis . The European Journal of Psychiatry. 2021; 2(002):1-12. https://doi.org/10.1016/j.ejpsy.2021.02.00

[6]O’ Connor RC, Kirtley OJ. The integrated motivational–volitional model of suicidal behaviour. Philosophical Transactions of the Royal Society B: Biological Sciences. 2018; 373(1754):20170268. http://doi:10.1098/RSTB.2017.0268 PMID:30012735

[7]O’ Connor RC, Smyth R, Ferguson E, Ryan C, Williams JM. Psychological processes and repeat suicidal behavior: a four- year prospective study. Journal of Consulting and Clinical Psychology. 2013; 81(6):1137-1143. http://doi:10.1037/a0033751 PMID:23855989

Piont 3: Introduction, line 151, should be a new paragraph as does not belong to subheading 1.3

Response 3: As the Reviewer pointed out, the content of lines 161-170 does not really belong to 1.3, so we have added a subheading "1.4. Research hypotheses" in the text according to your suggestion. We also feel that it would be more appropriate to use this subsection to present all the assumptions of this study.

Piont 4: Introduction, lines 154-155 Hypothesis 1, this has already been tested, see: Leung, C.L.K., Kwok, S.Y.C.L. & Ling, C.C.Y. An Integrated Model of Suicidal Ideation in Transcultural Populations of Chinese Adolescents. Community Ment Health J 52, 574–581 (2016). https://doi.org/10.1007/s10597-015-9920-2. Therefore, please adjust to provide a rationale for why you are testing this in this population and cite this paper in the intro

Response 4: We are particularly grateful to the Reviewer for providing the literature, which has allowed us to further enrich and expand the evidence for our study.

At first, it is really true as Reviewer suggested that, a cross-sectional study by Leung et al. [1] of adolescents aged 11 to 18 years in two Chinese developed cities, Hong Kong and Shanghai, confirmed that family functioning significantly predicted suicidal ideation. As suicide is a social and cultural phenomenon, the study in a sample of two developed cities in China, although well representative, with the changing economic and social patterns and the far-reaching impact of the global COVID-19 pandemic, the suicide of adolescents in more and more widely developing cities and inland areas of China awaits in-depth research and exploration of its intervention options. As our study data were obtained from a large prevalence survey across four provinces and cities in China, the finding is representative and has certain implications for a systematic understanding of how family functioning affects adolescent suicidal ideation, and for the development of effective suicide prevention measures for Chinese adolescents by educational authorities and mental health agencies.

Secondly, there are many influencing factors and theoretical models of suicide ideation. The study by Leung et al. integrated the Family Ecological System Theory and the Diathesis-Stress-Hopelessness Model of suicidal ideation in connecting the correlates. The results supported an integrative approach in facilitating parent-adolescent communication and strengthening family functioning, and reducing the use of negative social problem-solving styles in adolescent suicide prevention. But our study was based on the Ecological System Theory and the IMV Model. The IMV Model is the representative and most widely used theoretical model in the studies of adolescent suicide. The results of our study proved that the relevant variables of this model, such as family functioning, defeat and meaning in life, play a significant role in the process of adolescent suicidal ideation. It provided a practical reference for the comprehensive psycho-social intervention model of adolescent suicidal behavior from a different perspective.

Piont 5: Methods as heading missing, which is standard; remove Subjects and measures

Response 5: As Reviewer suggested that, we have re-titled the section as follows:

     2. Participants and measures

     2.1. Participants

     2.2. Measures

     2.3. Statistical Analysis

Piont 6: Discussion - it is missing how the results are useful theoretically and practically, as cited in the conclusion. Only mention is line 437 section limitations: 'these findings are helpful for education departments and mental health institutions to devise effective suicide prevention plans for Chinese adolescents' which is not specific or informative enough.

Response 6: It is really true as the Reviewer suggested that, the presentation related to the application of the findings of this study in practice is relatively simple and not specific enough. Based on the comments of the Reviewer and further review of the literature, we have made great changes to the discussion part, specifically increased the content of the practical value as “5. Practical Significance” of our research conclusions for the prevention and intervention of adolescent suicide in China. We particularly elaborated the need for the implementation of psycho-social intervention to prevent suicide in adolescents. The changes can be seen in line 415-457 of the revised version of the paper. The references are as follows:

[1]Bartoli F, Cavaleri D, Moretti F, Bachi B, Calabrese A, Callovini T, Cioni RM, Riboldi I, Nacinovich R, Crocamo C, Carrà G. Pre-Discharge Predictors of 1-Year Rehospitalization in Adolescents and Young Adults with Severe Mental Disorders: A Retrospective Cohort Study. Medicina (Kaunas). 2020; 56(11):613. http://doi: 10.3390/medicina56110613 PMID: 33203127

[2]Calear AL, Christensen H, Freeman A, Fenton K, Busby Grant J, van Spijker B, Donker T. A systematic review of psychosocial suicide prevention interventions for youth. Eur Child Adolesc Psychiatry. 2016; 25(5):467-82. http://doi:10.1007/s00787-015-0783-4. Epub 2015 Oct 15 PMID: 26472117

[3]Díaz-Oliván A, Porras-Segovia ML, Barrigón L, Jiménez-Muñoz E, Baca-García. Theoretical models of suicidal behaviour: A systematic review and narrative synthesis. The European Journal of Psychiatry. 2021; 2(002):1-12. https://doi.org/10.1016/j.ejpsy.2021.02.00

[4]Zhang N, Yuan B, Wang K, Shen T. The influence of impulsive traits on high school students' suicidal ideation: the role of campus rejection and sense of meaning in life. Psychological and Behavioral Research. 2021; (01):89-95

[5]Erikson EH. Identity: Youth and crisis. New York: Norton. 1968

[6]Fitzpatrick JJ., Preventing suicide: developing meaning in life. Arch Psychiatr Nurs. 2009; 23(4): 275-276. http://doi:10.1016/j.apnu.2009.06.002 

[7]Frankl VE. Man’s search for meaning: an introduction to logotherapy. Oxford: Washington Square Press. 1963. http://doi:10.1016/j.jebo.2008.01.004

We tried our best to improve the manuscript. For example, we have further streamlined the length of this paper, refined the expression of words and verified the accuracy of the data analysis. These changes will not influence the content and framework of this paper. And here we did not list the changes completely but marked in yellow in revised paper.

We appreciate for the Reviewer’ warm work earnestly, and hope that the correction will meet with approval. If there’s any further questions or comments please let us know.

Once again, thank you very much for your comments and suggestions.

Best wishes,

Qin Yang 

Reviewer 2 Report

The submission I had the opportunity to review reports a study about the link between family functioning and suicidal ideation in Chinese adolescents, a topic which should be of interest to all professionals involved in adolescents’ mental healthcare.

The study seems to have been conducted properly, and the article is well written. Analyses are thorough and results are reported precisely (although, since I am not a statistician, I would suggest a check from an expert). In their Discussion, the Authors are able to provide a nice explanation and contextualization of their findings.

Nevertheless, some points can be improved to further increase the value of this article. Please find my comments hereafter.

1. The Introduction section is somewhat too long, and in some passages (e.g., LL 108-115) it sounds like a mere list of the existing evidence that affects flow and readability. The Authors should try to do a more coherent synthesis.

2. The description of study sample, being a result of this research, should be moved from the Methodssection (LL 172-180) to the Results section, following the STROBE statement (https://www.strobe-statement.org).

3. In the Limitations and future implications section, the Authors state that their “findings are helpful for education departments and mental health institutions to devise effective suicide prevention plans for Chinese adolescents”. However, they come short of further discussing implications in deep. Given the relevance of this matter, I would suggest that the Authors discuss the possible clinical implications of their research in terms of patients’ management, such as:

3a. the increased burden caused by suicidality in adolescents, not only for the patient but also for mental health services (Bartoli et al., 2020. https://doi.org/10.3390/medicina56110613);

3b. the need for the implementation of psychosocial interventions to prevent suicide in adolescents (Calear et al., 2020, https://doi.org/10.1007/s00787-015-0783-4).

4. The sentence “The Cronbach's α coefficient of the scale was 0.917 in this study” (LL 197-198) is repeated twice.

5. Some typos and editing inaccuracies should be corrected.

Author Response

Response to Reviewer 1 Comments

Dear Reviewer:

We are very grateful to your comments for the manuscript. Those comments are all valuable and very helpful for revising and improving our paper, as well as the important guiding significance to our researches. We have studied comments carefully and have made correction which we hope meet with approval. Revised portion are marked in red in the paper. The main corrections in the paper and the responds to the reviewer’s comments are as flowing:

Responds to the Reviewer’s comments:

Piont 1: The Introduction section is somewhat too long, and in some passages (e.g., LL 108-115) it sounds like a mere list of the existing evidence that affects flow and readability. The Authors should try to do a more coherent synthesis.

Response 1: As suggested by the Reviewer, we have re-examined the content of the full text, made some changes to the content of the introduction and discussion section, mainly including the accuracy of language expression and the deletion or replacement of lengthy expressions. We have marked the revised part in yellow in the revised draft. Please review and further guidance.

Piont 2: The description of study sample, being a result of this research, should be moved from the Methodssection (LL 172-180) to the Results section, following the STROBE statement (https://www.strobe-statement.org).

Response 2: Thanks the Reviewer’s suggestion. According to the contribution template of “International Journal of Environmental Research and Public Health”, we introduced the characteristic description of the sampled objects in the part of the research method. We also studied the recent papers published in the magazine and found that there was a similar practice. The references are as follows:

[1]Figueiredo S, Vieira R. The Effect of Chronotype on Oppositional Behaviour and Psychomotor Agitation of School-Age Children: A Cross-Sectional Study. Int. J. Environ. Res. Public Health. 2022; 19:2-17. https://doi.org/10.3390/ ijerph192013233

[2]Jędryczka W, Sorokowski P, Dobrowolska M. The Role of Victim's Resilience and Self-Esteem in Experiencing Internet Hate. Int J Environ Res Public Health. 2022; 19(20):13149. http://doi: 10.3390/ijerph192013149

Piont 3: In the Limitations and future implications section, the Authors state that their “findings are helpful for education departments and mental health institutions to devise effective suicide prevention plans for Chinese adolescents”. However, they come short of further discussing implications in deep. Given the relevance of this matter, I would suggest that the Authors discuss the possible clinical implications of their research in terms of patients’ management, such as:

3a. the increased burden caused by suicidality in adolescents, not only for the patient but also for mental health services (Bartoli et al., 2020. https://doi.org/10.3390/medicina56110613);

3b. the need for the implementation of psychosocial interventions to prevent suicide in adolescents (Calear et al., 2020, https://doi.org/10.1007/s00787-015-0783-4).

Response 3: We are particularly grateful to the Reviewer for providing the literature, which has allowed us to further enrich and expand the evidence for our study. It is really true as the Reviewer suggested that, the presentation related to the application of the findings of this study in practice is relatively simple and not specific enough. Based on the comments of the Reviewer and further review of the literature, we have made great changes to the discussion part, specifically increased the content of the practical value as “5. Practical Significance” of our research conclusions for the prevention and intervention of adolescent suicide in China. We particularly elaborated the need for the implementation of psycho-social intervention to prevent suicide in adolescents. The changes can be seen in line 415-457 of the revised version of the paper. The references are as follows:

[1]Bartoli F, Cavaleri D, Moretti F, Bachi B, Calabrese A, Callovini T, Cioni RM, Riboldi I, Nacinovich R, Crocamo C, Carrà G. Pre-Discharge Predictors of 1-Year Rehospitalization in Adolescents and Young Adults with Severe Mental Disorders: A Retrospective Cohort Study. Medicina (Kaunas). 2020; 56(11):613. http://doi: 10.3390/medicina56110613 PMID: 33203127

[2]Calear AL, Christensen H, Freeman A, Fenton K, Busby Grant J, van Spijker B, Donker T. A systematic review of psychosocial suicide prevention interventions for youth. Eur Child Adolesc Psychiatry. 2016; 25(5):467-82. http://doi:10.1007/s00787-015-0783-4. Epub 2015 Oct 15 PMID: 26472117

[3]Díaz-Oliván A, Porras-Segovia ML, Barrigón L, Jiménez-Muñoz E, Baca-García. Theoretical models of suicidal behaviour: A systematic review and narrative synthesis. The European Journal of Psychiatry. 2021; 2(002):1-12. https://doi.org/10.1016/j.ejpsy.2021.02.00

[4]Zhang N, Yuan B, Wang K, Shen T. The influence of impulsive traits on high school students' suicidal ideation: the role of campus rejection and sense of meaning in life. Psychological and Behavioral Research. 2021; (01):89-95

[5]Erikson EH. Identity: Youth and crisis. New York: Norton. 1968

[6]Fitzpatrick JJ., Preventing suicide: developing meaning in life. Arch Psychiatr Nurs. 2009; 23(4): 275-276. http://doi:10.1016/j.apnu.2009.06.002 

[7]Frankl VE. Man’s search for meaning: an introduction to logotherapy. Oxford: Washington Square Press. 1963. http://doi:10.1016/j.jebo.2008.01.004

Piont 4: The sentence “The Cronbach's α coefficient of the scale was 0.917 in this study” (LL 197-198) is repeated twice.

Response 4: We are very sorry for our repetitive writing content. We have made correction according to the Reviewer’s comments.

Piont 5: Some typos and editing inaccuracies should be corrected.

Response 5: We are very sorry for our typos and editing inaccuracies. We have re-written this part according to the Reviewer’s suggestion. We tried our best to improve the manuscript. For example, we have further streamlined the length of this paper, refined the expression of words and verified the accuracy of the data analysis. These changes will not influence the content and framework of this paper. And here we did not list the changes completely but marked in yellow in revised paper.

We appreciate for the Reviewer’ warm work earnestly, and hope that the correction will meet with approval. If there’s any further questions or comments please let us know.

Once again, thank you very much for your comments and suggestions.

Best wishes,

Qin Yang 

Round 2

Reviewer 2 Report

I thank the Authors for having taken into account my suggestions.